# Monitoring Scheme for the Detection of Hydrogen Leakage from a Deep Underground Storage. Part 1: On-Site Validation of an Experimental Protocol via the Combined Injection of Helium and Tracers into an Aquifer

**Stéphane Lafortune [1,\*], Philippe Gombert [1] , Zbigniew Pokryszka [1], Elodie Lacroix [1,2], Philippe de Donato [2] and Nevila Jozja [3]**

[1] Ineris, Parc Technologique Alata, 60550 Verneuil-en-Halatte, France; philippe.gombert@ineris.fr (P.G.); zbigniew.pokryszka@ineris.fr (Z.P.); elodie.lacroix@ineris.fr (E.L.)

[2] GéoRessources Laboratoire, Université de Lorraine-CNRS, 54500 Vandoeuvre-lès-Nancy, France; philippe.de-donato@univ-lorraine.fr

[3] Ecole Polytechnique, Laboratoire CETRAHE, Université d'Orléans, 8 rue Léonard de Vinci, 45100 Orléans, France; nevila.jozja@univ-orleans.fr

\* Correspondence: stephane.lafortune@ineris.fr; Tel.: +33-(0)-344-556-791

**Abstract:** Massive underground storage of hydrogen could be a way that excess energy is produced in the future, provided that the risks of leakage of this highly flammable gas are managed. The ROSTOCK-H research project plans to simulate a sudden hydrogen leak into an aquifer and to design suitable monitoring, by injecting dissolved hydrogen in the saturated zone of an experimental site. Prior to this, an injection test of tracers and helium-saturated water was carried out to validate the future protocol related to hydrogen. Helium exhibits a comparable physical behavior but is a non-flammable gas which is preferable for a protocol optimization test. The main questions covered the gas saturation conditions of the water, the injection protocol of 5 m$^3$ of gas saturated water, and the monitoring protocol. Due to the low solubility of both helium and hydrogen, it appears that plume dilution will be more important further than 20 m downstream of the injection well and that monitoring must be done close to the well. In the piezometer located 5 m downstream the injection well, the plume peak is intended to arrive about 1 h after injection with a concentration around 1.5 mg·L$^{-1}$. Taking these results into account should make it possible to complete the next injection of hydrogen.

**Keywords:** hydrogen; underground storage; leakage; monitoring; protocol; helium; aquifer

## 1. Introduction

### 1.1. General Information Regarding the Underground Storage of Hydrogen

To contribute more effectively to the fight against climate change and the preservation of the environment, as well as reinforcing their energy independence, France published the Energy transition law for green growth in 2015 [1]. This law aims to increase the share of renewable energies to 23% of gross final energy consumption in 2020 and 32% in 2030, compared to 16% currently [2]. The development of these renewable energies will come up against the need to manage the fluctuating or intermittent nature of some of them. This will involve storing the energy produced in excess or not consumed so that this energy can be re-used later (directly as fuel or mixed with natural gas,

or indirectly by converting it into heat or electricity). The underground environment has many advantages with regard to its potential for high capacity storage in the short or medium-term [3]. France already has 100 operational underground reservoirs of which 78 are salt cavities: these are very large underground cavities, of the order of a million $m^3$, formed by injecting freshwater into deep salt formations. Currently, the storage capacity of all of these salt cavities together totals around 14 million $m^3$ of liquid or liquefied hydrocarbons and 2 billion $m^3$ of natural gas [3]. Against a background of the gradual abandonment of fossil fuels, a number of research studies are looking into the possibility of storing hydrogen ($H_2$) in such deep salt cavities in the future.

It is within this context that the ROSTOCK-H project (Risks and Opportunities of the Geological Storage of Hydrogen in Salt Caverns in France and Europe) has been financed by GEODENERGIES the French Scientific Interest Group. This project started in 2017 and will end in 2021. One of its objectives is to define monitoring methods for the detection of sub-surface hydrogen leakage, with the dual aim of (i) sizing a measurement scheme capable of detecting diffuse hydrogen leaks, and (ii) studying the diffusive process and the chemico-physical impacts of hydrogen in a shallow aquifer. The approach is centered on two experimental simulations separated in time on the same experimental site. Simulation 1 consists of analyzing the migration in groundwater of a plume of water saturated with neutral gas (helium) and containing various tracers. The objective is to test the operation protocol envisaged for the future injection of hydrogen and to optimize the associated monitoring systems. Simulation 2 consists of creating a plume of dissolved hydrogen in groundwater, according to the same protocol used with helium, to simulate a sudden and brief leak from a deep geological hydrogen storage site towards a shallow aquifer. The evolution of the plumes thus created in the saturated zone, and any potential outgassing to non-saturated zone and the surface will then be followed. All of these simulations will take place at the Catenoy (Northern France) experimental site, which has already been used in the context of similar experiments that studied the behavior of $CO_2$ for the purpose of Carbon Capture and Storage, or CCS [4,5].

The first injection simulation, which is the subject of this article, therefore involves helium and aims to size the entire leak simulation system and to adapt its protocol and monitoring for the simulation of a hydrogen leak which will subsequently be carried out on the same experimental site. Helium, the gas is chosen for this test, exhibits a physical behavior similar to that of hydrogen, in particular a very low solubility and a high diffusion coefficient in water. At the same time, it is a non-flammable gas, as opposed to the highly flammable hydrogen. This fact makes the organization of this pre-test less complicated from a safety point of view while respecting the similarities with the future hydrogen experiment.

The test site is located in the chalk layer within the Paris Basin. The protocol adopted consists of extracting water from the shallow aquifer, saturating it with gas (helium), and then reinjecting it into the aquifer with tracers to follow the propagation of the dissolved gas plume. This test aims to improve the experimental protocol to be used for the subsequent experiment involving injecting dissolved hydrogen into the aquifer (simulation no. 2).

*1.2. Risks Associated with Underground Hydrogen Storage*

If there is a leak coming from a deep geological reservoir, the gas will migrate to the surface. In most cases, it will encounter at least one aquifer before reaching the surface [6]. If the leakage rate exceeds the dissolution potential of hydrogen in the groundwater, which is of the order of 2 mg·$L^{-1}$ at surface conditions, which is low compared to other gases (11 mg·$L^{-1}$ for dioxygen, 24 mg·$L^{-1}$ for dinitrogen, 2500 mg·$L^{-1}$ for carbon dioxide), part of the hydrogen will continue its migration to the surface. Hydrogen is then likely to accumulate in a confined underground area near the reservoir (cellar, underground car park, urban underground network, tunnel, etc.) where it will become a risk factor for explosion, fire, or asphyxiation. Indeed, hydrogen is a highly flammable gas with a very wide explosive range of between 4% and 75% at ambient pressure and temperature [7].

In the event of a potential hydrogen leak, the aquifer, therefore, represents the last warning barrier on the path of migration to the surface [6]. By transporting information from upstream to downstream, the aquifer constitutes a very favorable environment for the implementation of an integrated monitoring system immediately downstream of a deep storage site. As dissolved hydrogen is not normally present in water, detecting it within an aquifer will indicate a potential leak. This could be manifested as a direct detection ($H_2$ dissolved in water) or indirectly by means of the effects caused by this strongly reducing gas: decrease of the oxidation-reduction potential, decrease in the content of other dissolved gases in the water (mainly $N_2$, $O_2$, and $CO_2$), and oxidation-reduction reactions, for example [8–14]:

- reduction of nitrates ($NO_3^-$) to nitrites ($NO_2^-$), or even to ammonium ($NH_4^+$), and then to gaseous nitrogen ($N_2$);
- reduction of sulfates ($SO_4^{2-}$) to sulfides ($SO_3^-$), or even to hydrogen sulfide ($H_2S$);
- reduction of iron III to iron II;
- dissolving of metallic trace elements, if they are present in the aquifer rock, following the lowering of the oxidation-reduction potential.

The literature shows that, under normal pressure and temperature conditions, the reduction of nitrates and sulfates cannot take place except in the presence of a catalyst such as iron, copper, nickel, or platinum [8–14]. However, the frequent use of stainless steel, which contains iron and a significant amount of nickel (up to 20%), in the metal casings of a large number of water boreholes (for drinking water, mineral water, etc.) and hydrocarbon wells inevitably brings some of these catalysts into contact with the groundwater.

## 2. Materials and Methods

### 2.1. Presenting the Catenoy Site

The Catenoy (Oise) experimental site is located about 50 km north of Paris in the Paris sedimentary basin (Figure 1). The coordinates of the center of the site are as follows: latitude 49°22′05″ N, longitude 2°30′26″ E, altitude ~60 m asl. The geology corresponds to a few meters of Quaternary deposits (colluviums, loess) and Tertiary formations, lying over a hundred meters of Senonian chalk that is only visible in the thalwegs (see also Figure 2b). Under the site, the underground geology of the first 25 m is 3 m of colluvium, 4 m of Thanetian sands, and 18 m of chalk. At a few hundred meters on the south of the site, Ypresian and Lutetian layers form a hill. There are no major tectonic accidents nearby and the bedding of the chalk formation is horizontal. This chalk encloses an aquifer with a static level at a depth of 13 m which flows in the WSW-ENE direction [4].

Located in a former agricultural field that has been fallow for more than a decade, the site is equipped with 8 piezometers aligned in the direction of flow of the aquifer over a distance of 80 m, referenced PZ1 to PZ6 (including supplementary piezometers PZ2BIS and PZ2TER), plus a technical shed housing the monitoring material (Figure 2). The piezometers are about 25 m depth and are screened in the aquifer starting from 13 m depth.

The hydrodynamic characteristics of the chalk aquifer at the site were determined during a pumping test carried out in 2013 [4]. Depending on the piezometer considered, the following values were reported:

- Storage coefficient (porosity): $1.1 \times 10^{-2}$ to $6.5 \times 10^{-2}$
- Hydraulic conductivity (permeability): $6.4 \times 10^{-4}$ to $1.4 \times 10^{-3}$ m·s$^{-1}$

In addition, a previous tracing test made it possible to estimate the flow rate of the aquifer at 3 m·d$^{-1}$ at PZ3 and PZ5, and 10 m·d$^{-1}$ at PZ4, the latter being situated in a preferential flow path (fissured area). This test, therefore, demonstrates the double porosity of the aquifer studied.

There is also a meteorological station on-site to measure the following parameters at hourly time intervals: atmospheric temperature and pressure, rainfall.

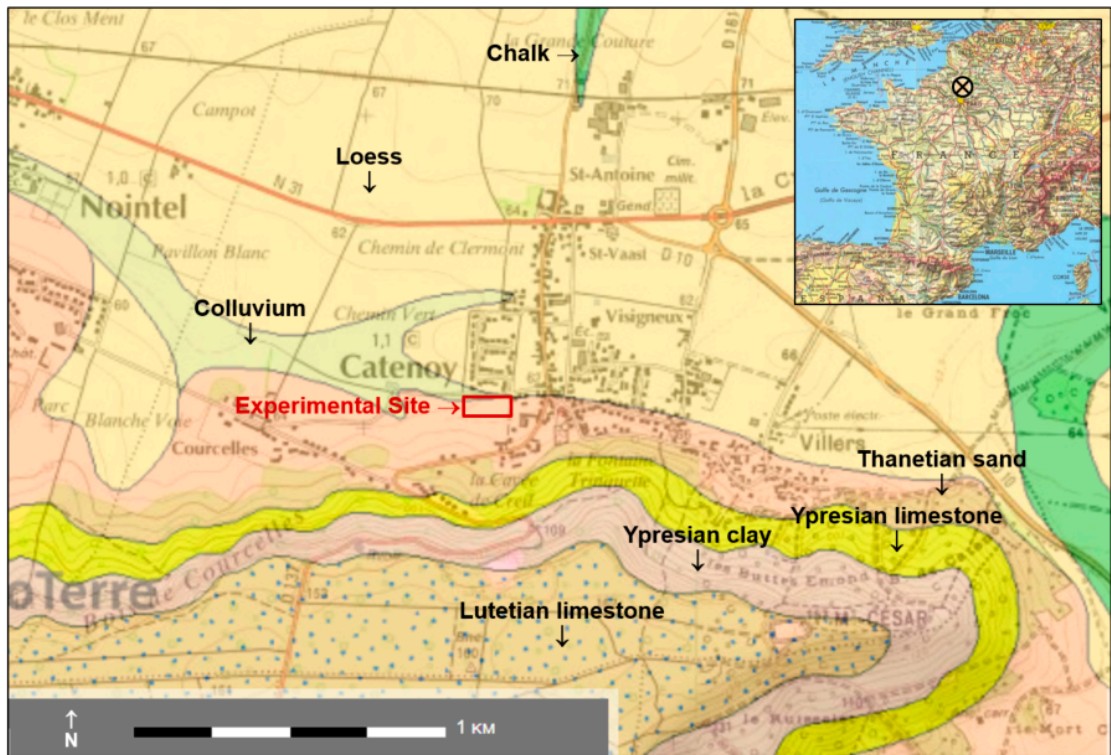

**Figure 1.** Location and geological context of the experimental site.

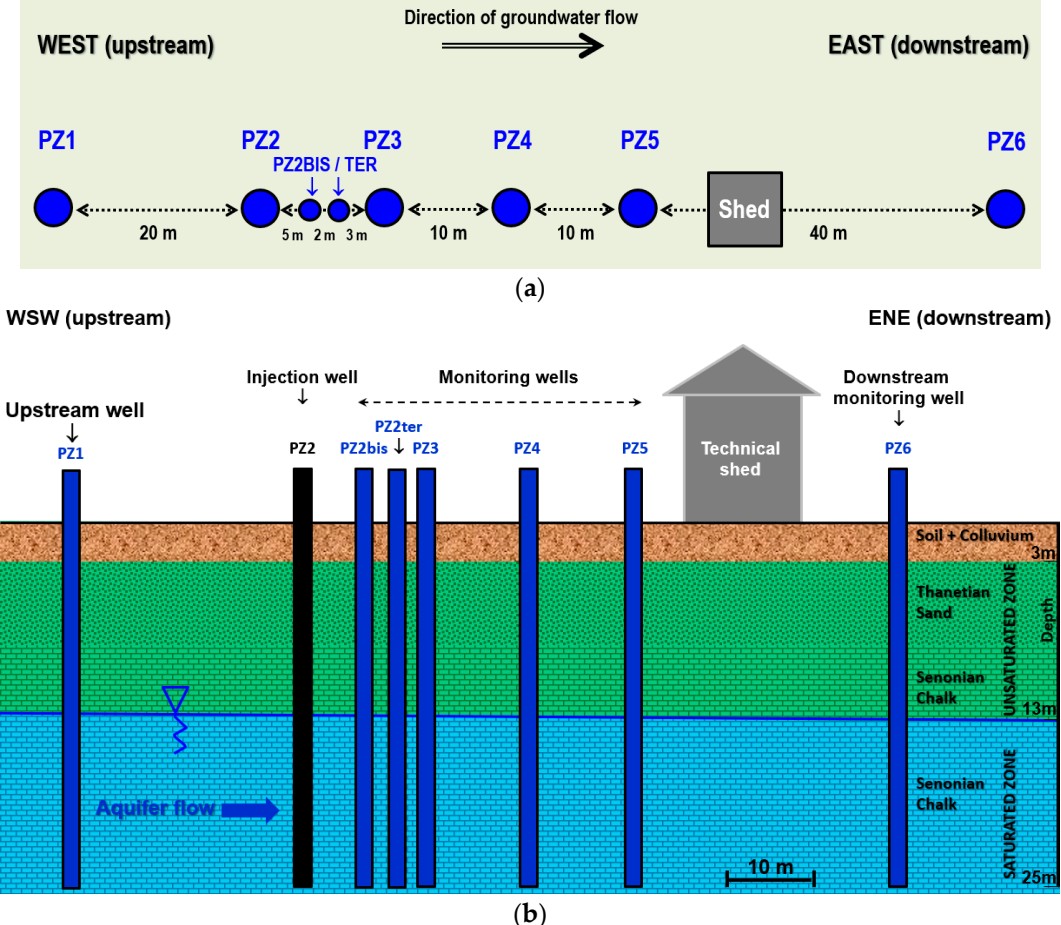

**Figure 2.** Diagram of the Catenoy experimental site: (**a**) planimetry; (**b**) cross-section.

## 2.2. Establishing the Baseline

Hydrogen is a very mobile gas that can leak towards the surface and accumulate in the groundwater and the soil. A complete monitoring protocol could interest the saturated zone, the unsaturated zone, the soil, and the surface because hydrogen can be detected in all these compartments as it can be seen in natural hydrogen emission areas [15]. However, the leakage simulation protocol (see under) is based on the injection of water saturated with dissolved gas (helium or hydrogen) directly into the aquifer. The water will be previously saturated at atmospheric conditions, i.e., at a pressure of 0.10 MPa, and then injected from 2 m to 11 m under the water table, where the hydrostatic pressure is between 0.12 and 0.21 MPa. Thus, the water will always remain undersaturated and nor helium nor hydrogen will degas. In these conditions, the only way for the dissolved gas to propagate is to follow the groundwater flow. In this study, the monitoring system has thus be designed for the saturated zone, with a light control of eventual weak degassing in the internal atmosphere of the piezometers but only for safety purposes.

Prior to setting up a monitoring system for the hydrogen injection test, a baseline of the initial piezometric, chemico-physical and hydrogeochemical values of the aquifer was established over 388 days starting on 27 October 2018. On all of the 7 main piezometers of the site (PZ1, PZ2, PZ2BIS, PZ3, PZ4, PZ5, and PZ6), the baseline corresponds to more than 200 measurements of each of the main chemico-physical parameters of the water: pH, temperature, electrical conductivity (EC), oxidation-reduction potential (ORP) and dissolved $O_2$ (Table 1). The water has a neutral pH (pH = 7.25), is moderately mineralized (EC = 562 $\mu S \cdot cm^{-1}$), and oxygenated ($O_2$ = 5.44 $mg \cdot L^{-1}$) due to its proximity to the soil surface, and is thus globally oxidative (ORP = +103 mV).

**Table 1.** Baseline of the chemico-physical parameters.

| Parameter | $O_2$ | pH | T | EC | ORP |
|---|---|---|---|---|---|
| Unit | $(mg \cdot L^{-1})$ | - | (°C) | $(\mu S \cdot cm^{-1})$ | (mV) |
| Number | 208 | 223 | 224 | 223 | 221 |
| Average | 5.4 | 7.3 | 12.1 | 562 | 103 |
| SD | 1.7 | 0.3 | 0.6 | 66 | 89 |

Legend: $O_2$ = dissolved oxygen; T = Temperature; EC = Electric Conductivity; ORP = Oxidation-Reduction Potential; SD = Standard Deviation.

These chemico-physical parameters are quite stable over space and time. Figure 3 represents the boxplots of the dissolved oxygen and the oxidation-reduction potential at each piezometer during the baseline. Figure 4 represents the evolution of these main chemico-physical parameters over time and seems to show a certain sensitivity to the depth of the aquifer, which varies from 13.06 m to 13.94 m (Figure 3).

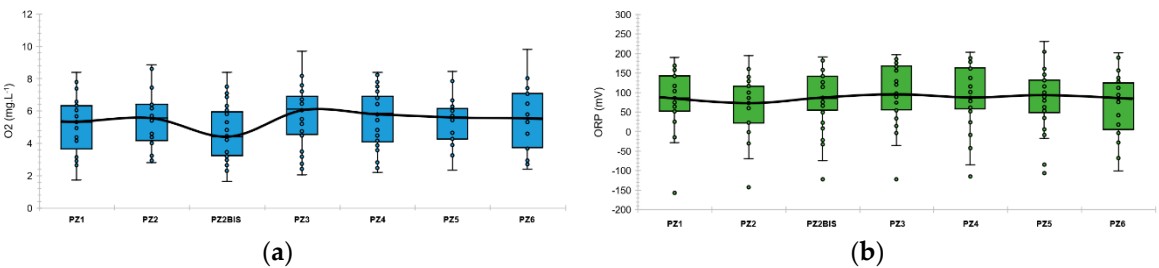

(**a**)  (**b**)

**Figure 3.** Boxplots of dissolved oxygen concentration and oxidation-reduction potential along the experimental site during the baseline: (**a**) Dissolved oxygen concentration; (**b**) oxidation-reduction potential. The colored boxes represent the 1st and 3rd quartiles (respectively Q1 and Q3), the black line is the median line, the dots are the measured values, the whiskers are the upper and lower extreme limits calculated according to the Tukey's formula: Q1 + 1.5 (Q3 − Q1) and Q3 − 1.5 (Q3−Q1).

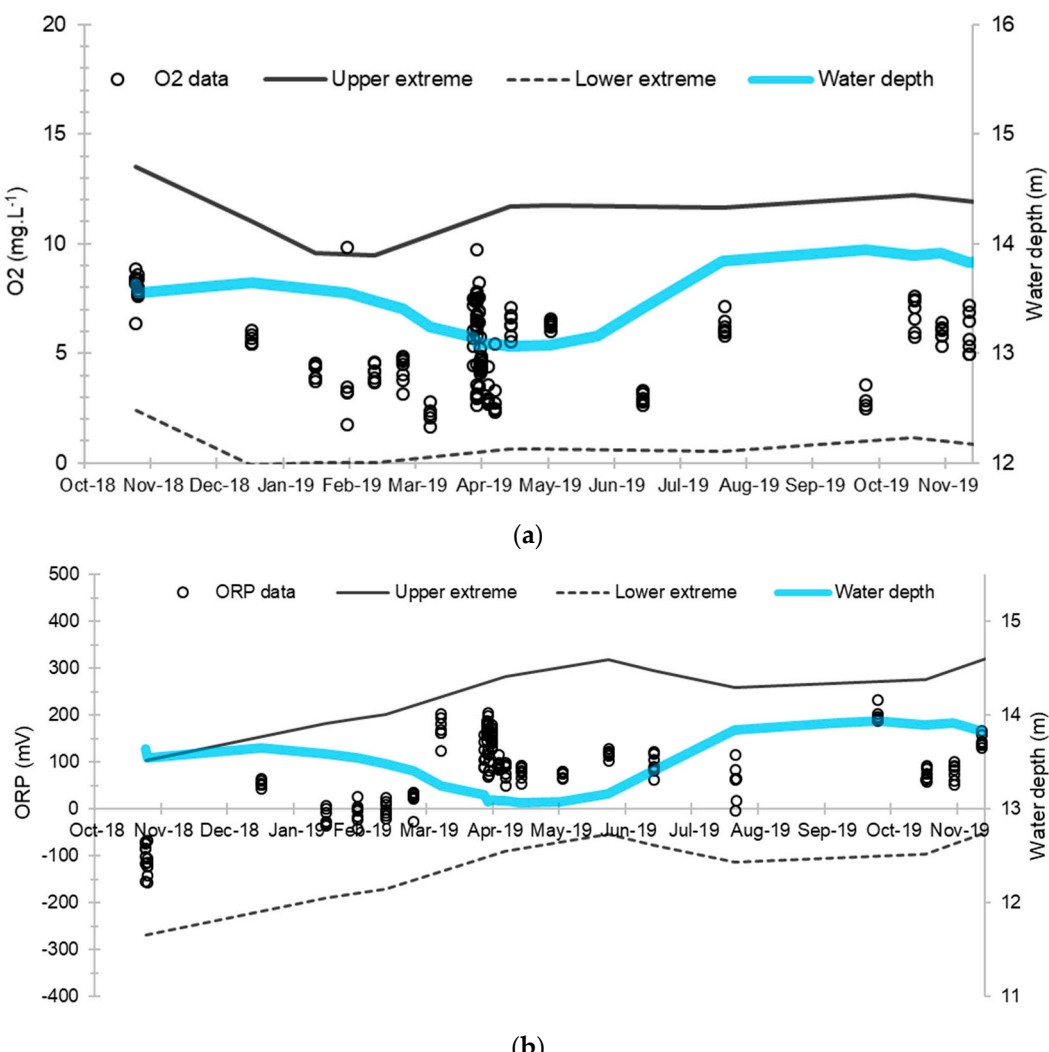

**Figure 4.** Evolution of dissolved oxygen concentration and oxidation-reduction potential and their extreme limits during the establishment of the baseline: (**a**) Dissolved oxygen concentration; (**b**) Oxidation-reduction potential. The solid and dashed lines correspond respectively to the upper and lower extreme limits calculated according to the Tukey's formula: Q1 + 1.5 (Q3 − Q1) and Q3 − 1.5 (Q3 − Q1) where Q1 and Q3 are the 1st and 3rd quartiles.

During the acquisition of the baseline data, 94 water samples were taken to analyze the major ions ($Ca^{2+}$, $Mg^{2+}$, $Na^+$, $K^+$, $HCO_3^-$, $Cl^-$, $SO_4^{2-}$, $NO_3^-$) and the main minor ions liable to be modified by a hydrogen-water-rock interaction ($NO_2^-$, $NH_4^+$, $SO_3^{2-}$, $S_2^-$, Fe, Mn). To comply with the storage conditions for all the elements, the analyses were carried out within 24 h of each sample being taken, using the methods presented in Table 2.

**Table 2.** Analytical methods and detection thresholds for the analyzed elements ($mg·L^{-1}$).

| Parameter | $HCO_3^-$ | $Ca^{2+}$ | $Mg^{2+}$ | $Na^+$ | $K^+$ | $Cl^-$ | $SO_4^{2-}$ | $NO_3^-$ | $NO_2^-$ | $NH_4^+$ | $SO_3^{2-}$ | $S_2^-$ | Fe | Mn |
|---|---|---|---|---|---|---|---|---|---|---|---|---|---|---|
| Method | Titration | | | | | Ionic Chromatography | | | | | | | ICP-MS | |
| DL ($mg·L^{-1}$) | 0.04 | | 0.05 | | | | 0.01 | 0.02 | 0.02 | | 0.01 | | 0.001 | |

Legend: ICP-MS = Inductively Coupled Plasma-Mass Spectrometry; IC = Ionic Chromatography; DL = Detection Limit.

Regarding the major elements, Table 3, Figures 5 and 6 show that their behavior is also very stable throughout the baselining. The water generally exhibits bicarbonate-calcium facies, characteristic of

chalk waters. This dominant hydrochemical facies is slightly altered by the presence of nitrate ions from agricultural inputs.

**Table 3.** Main characteristics of the major ions analyzed during baselining (mg·L$^{-1}$).

| Parameter | Ca$^{2+}$ | Mg$^{2+}$ | Na$^+$ | K$^+$ | HCO$_3^-$ | Cl$^-$ | SO$_4^{2-}$ | NO$_3^-$ | Total |
|---|---|---|---|---|---|---|---|---|---|
| Average | 97.1 | 11.5 | 12.6 | 4.69 | 298.8 | 23.6 | 27.9 | 33.4 | 97.1 |
| SD | 7.4 | 0.8 | 1.0 | 0.23 | 10 | 2.3 | 2.8 | 2.5 | 7.4 |

Legend: SD = Standard Deviation.

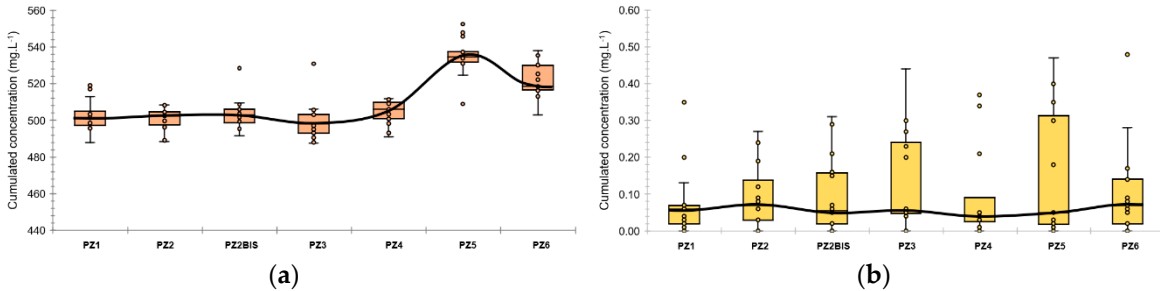

(a)  (b)

**Figure 5.** Boxplots of major and minor ions concentration along the experimental site during the baseline: (**a**) major elements; (**b**) minor elements. The colored boxes represent the 1st and 3rd quartiles (respectively Q1 and Q3), the black line is the median line, the dots are the measured values, the whiskers are the upper and lower extreme limits calculated according to the Tukey's formula: Q1 + 1.5 (Q3 − Q1) and Q3 − 1.5 (Q3 − Q1).

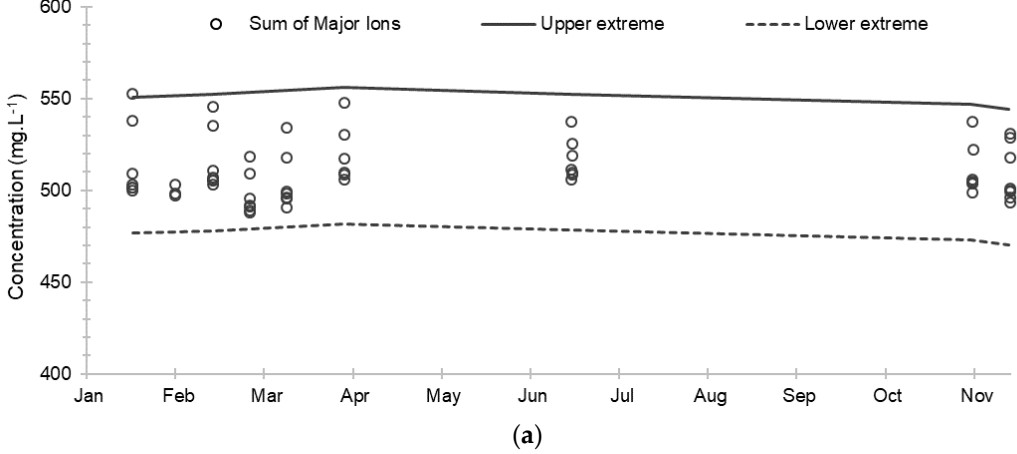

(a)

**Figure 6.** *Cont.*

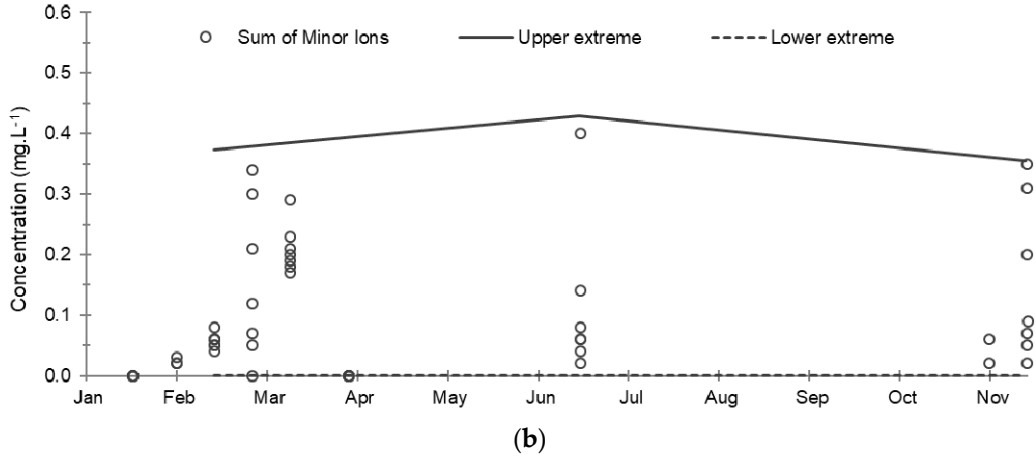

**Figure 6.** Evolution of the cumulative concentrations of major and minor ions and their extreme limits during the establishment of the baseline in 2019: (**a**) Sum of major ions; (**b**) Sum of minor ions. The solid and dashed lines correspond respectively to the upper and lower extreme limits calculated according to the Tukey's formula: Q1 − 1.5 (Q3 − Q1) and Q3 + 1.5 (Q3 − Q1) where Q1 and Q3 are the 1st and 3rd quartiles.

Regarding the minor elements analyzed, Table 4 shows the absence of nitrite and sulfide ions above the detection thresholds, as well the absence of sulfide ions except in five samples taken at PZ2 in the first half of 2019 where the concentrations ranged from 0.03 to 0.11 mg·L$^{-1}$. Ammonium ions are present in 74% of the samples, probably related to the application of nitrogenous fertilizers nearby, with a fairly fluctuating concentration with an average of 0.10 mg·L$^{-1}$. The totals of dissolved iron and manganese were also analyzed, but they were only minutely present in the water due to the mineralogical composition of the aquifer rock, which is made up of more than 95% calcite [4]: their ionized forms were therefore not researched. Ultimately, the evolution in the total of these minor ions varied little during baseline monitoring, the fluctuations observed being mainly due to the ammonium ions (Table 4).

**Table 4.** Main characteristics of minor ions analyzed during baselining.

| Parameter | $NO_2^-$ | $NH_4^+$ | $SO_3^{2-}$ | $S^{2-}$ | Total Fe | Total Mn | Total (N + S) |
|---|---|---|---|---|---|---|---|
| Average | <DL | 0.10 | 0.01 | <DL | 0.99 | 0.11 | 0.11 |
| SD | - | 0.13 | 0.03 | - | 1.58 | 0.22 | 0.13 |
| CV | - | 123% | (490%) | - | 160% | 193% | 114% |

Legend: DL = Detection Limit; SD = Standard Deviation; CV = Coefficient of Variation; Total (N+S) = Total of sulfur and nitrogen ions.

### 2.3. Preparing the Test

The helium was injected with the aim of testing and optimizing a future hydrogen injection device using an inert gas, and to configure the monitoring protocol (types of measurement and time intervals) depending on the piezometer being monitored. The objective of this test is to create a plume of dissolved helium in the aquifer, comparable to the future plume of dissolved hydrogen, and to monitor its propagation in the saturated zone.

Before this test, the propagation of the dissolved He plume was modeled in 1D using PHREEQC. Modeling parameters were determined using the results of previous $CO_2$ injection tests [4,5]. The result is shown in Figure 7 and shows a maximum dissolved helium concentration between 1.46 mg·L$^{-1}$ and $8 \times 10^{-21}$ mg·L$^{-1}$ from PZ2BIS to PZ6, and a peak arrival time between 100 min and 23 days. Peak values at PZ5 and PZ6 are expected to be below 1 μg·L$^{-1}$ and thus it will not be possible to detect helium in these two piezometers.

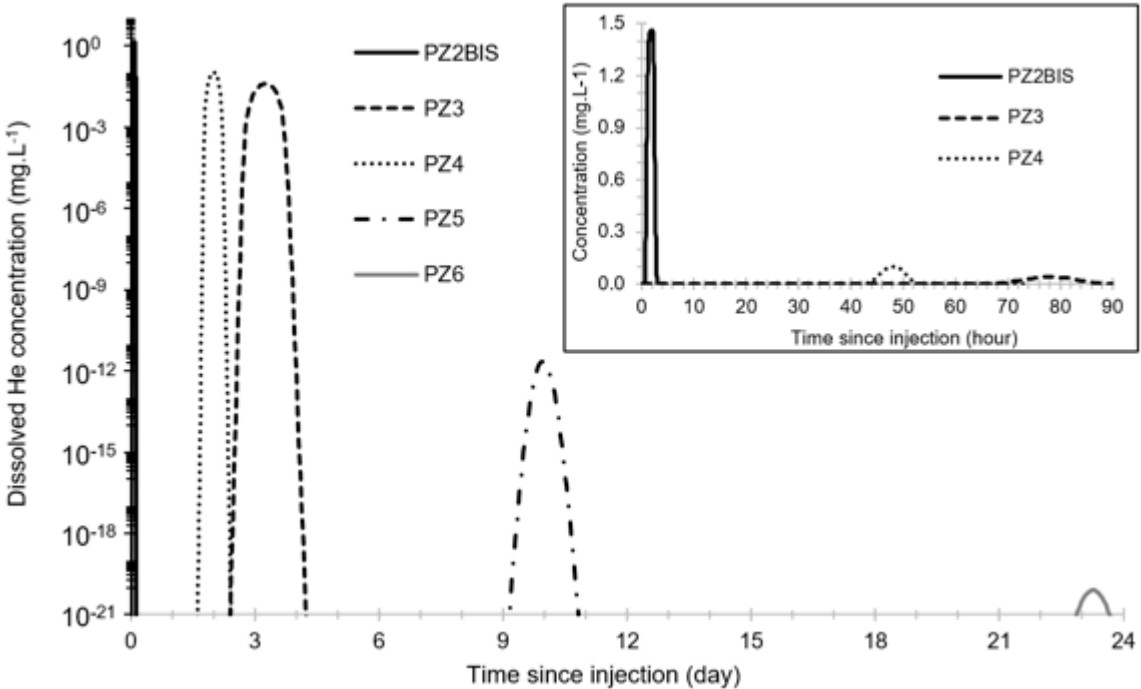

**Figure 7.** Model of the propagation of the dissolved He plume using PHREEQC.

Then, the water from the aquifer was extracted beforehand by pumping in the PZ2 piezometer (future injection well) to fill two HDPE tanks (Figure 8a): a first 1 m$^3$ tank which contains the tracers to help determine the arrival of the plume of dissolved gas and precisely quantify its kinetics, and a second 5 m$^3$ tank in which the water will be saturated with helium by bubbling it. It was decided not to incorporate the tracers in the tank of water saturated with helium to avoid the risk of reducing the solubility of this gas.

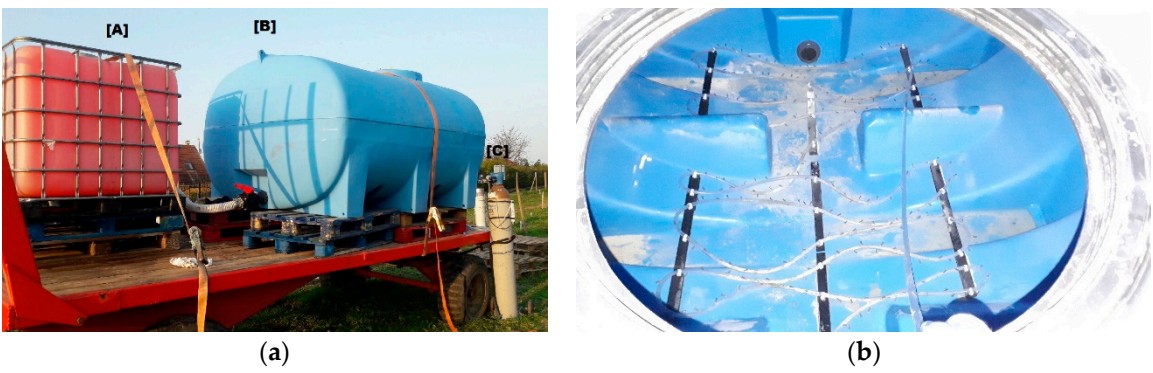

(**a**)                                                                 (**b**)

**Figure 8.** Views of the two tanks and the helium saturation device: (**a**) View of the 1 m$^3$ tracers tank [A], the 5 m$^3$ He saturated tank [B] and the compressed He cylinder [C]; (**b**) View from the manhole of the bubbling device in the 5 m$^3$ He tank (20 m of PVC pipe pierced with 200 needles of 0.5 mm diameter).

In the first 1 m$^3$ tank, two types of tracers were used:

- fluorescent organic tracers that exhibit no analytical interference between them, to allow in situ detection in real-time of the arrival of the injected plume thanks to the installation of a GGUN FL-30 field fluorimeter; these are uranine or fluorescein sodium (green dye), sulforhodamine B (red dye) and Amino G Acid (a colorless tracer emitting in blue); however, previous experiments with these types of organic tracers with a long carbonaceous molecule ($C_{20}$ to $C_{40}$) have shown

that not all of them were conservative when transferred to an aquifer composed of chalk with finely porous matrix permeability, as is the case in Catenoy [5];

- inorganic ionic tracers, which are highly conservative but colorless; they are analyzed a posteriori in the laboratory, from a water sample to precisely quantify the kinetics of the plume; these are lithium (as lithium chloride, LiCl) and bromide (as potassium bromide, KBr)

For the final hydrogen injection experiment, only the most efficient fluorescent tracer and ionic tracer in terms of their recovery will be used. The objective of this first test is, therefore, both to select these two tracers from the five tested, and to validate the principle of a prior injection of tracers to predict the arrival of the dissolved gas plume and, as a result, to improve the monitoring system. A quantity of 1 g of each tracer was diluted in the 1 m$^3$ tank: it will be noted that, for ionic tracers, this is 1 g of tracer ion (Li$^+$ and Br$^-$), which corresponds to 6.14 g of LiCl and 1.49 g of KBr.

To saturate the water with helium, a bubbling device was installed on the interior floor of the second tank to create a curtain of bubbles facilitating the dissolution of the gas. It is a PVC pipe of 20 m length, pierced with 200 holes (Figure 8b). This device is connected in a loop to a rotameter to regulate the flow of gas injected from a compressed gas cylinder (Figure 8a).

The two tanks were then emptied successively by gravity into the PZ2 borehole and the water then entered the aquifer.

The injection device consists of a reinforced PVC pipe 70 mm in diameter. This pipe is directly connected, by means of a tee fitting, to the outlet located at the base of the two tanks, each being isolated by a valve (Figure 9a). The injection pipe is plugged at its lower end and ballasted to ensure that it descends to the bottom of the well, at a depth of 25 m. To better distribute the injected fluid over the entire screened height in the injection well, the submerged part of the injection pipe is drilled with 46 holes that are distributed two by two from 12 to 23 m deep, and four by four from 23 to 25 m deep (Figure 9b). The shallowest holes are located 0.2 m below the water table to ensure that the dissolved helium is injected under a slight hydrostatic overpressure, and therefore cannot degas.

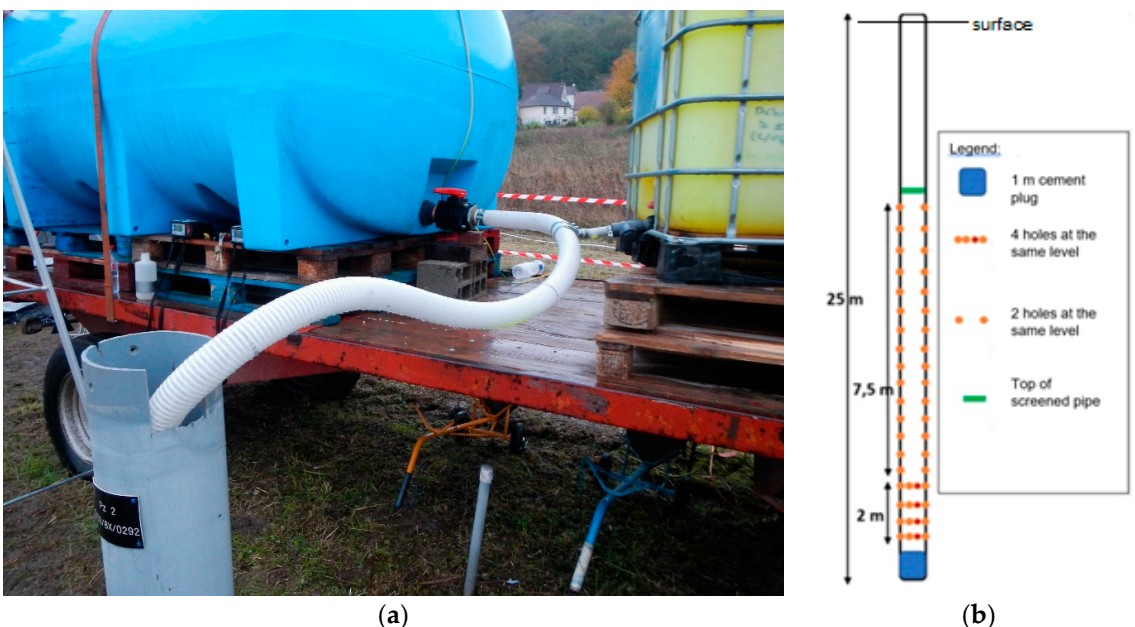

(**a**)    (**b**)

**Figure 9.** View of injection installation: (**a**) injection pipe and its connection to the tanks; (**b**) diagram of the arrangement of the holes along the injection pipe.

*2.4. Conducting the Test*

The tanks were filled with groundwater on the morning of 1 April 2019 using a submerged electric pump installed in PZ2 piezometer, the future injection well:

- the first 1 m$^3$ tank was filled with groundwater to 0.8 m$^3$, then the tracers, some of which are photosensitive, were added after sunset to avoid degradation by light; the volume of water was then increased to 1 m$^3$ for better mixing;
- the second 5 m$^3$ tank was filled with groundwater, then helium gas was continuously sent into the bubbling circuit until the following day at 12:00; the total time allowed for the helium to dissolve in the water was approximately 20 h; based on similar experiments carried out in the past, this was more than sufficient to ensure helium saturation of the water in the tank.

The injection of the water and tracers from the first tank (1 m$^3$) was performed by gravity on the 2nd of April 2019 from 10:35 to 11:10, which represents an injection flowrate of 1.7 m$^3 \cdot$h$^{-1}$. The helium-saturated water from the second tank (5 m$^3$) was injected again by gravity immediately after, i.e., from 11:12 to 12:47. This injection lasted 1 h 35 min, which corresponds to a flow rate of 3.2 m$^3 \cdot$h$^{-1}$.

### 2.5. Monitoring the Saturated Zone

The equipment installed to monitor the saturated zone during the helium injection test was as follows:

- Two physicochemical sensors for measuring temperature, pH, electrical conductivity, oxidation-reduction potential, and dissolved oxygen; one was permanently installed (for the duration of the test) in the PZ2BIS piezometer located 5 m downstream of the injection borehole while the other one was mobile to take measurements in all other piezometers;
- A GGUN-FL30 field fluorimeter which provides live analysis of the fluorescence of the water extracted from the piezometers; it is a multichannel device that can successively analyze the three fluorescent tracers used;
- A GRUNDFOS-MP1 submersible pump which was first used to fill the tanks from the PZ2 and moved to the PZ2BIS shortly before the start of the injection; it made it possible to regularly sample the groundwater to carry out laboratory analyses of the tracers, dissolved gases (helium) and major elements (calcium, magnesium, sodium, potassium, bicarbonates, chlorides, sulfates, nitrates);
- A Raman and Infrared (IR) spectrometer, installed in the PZ2TER piezometer located 7.5 m downstream of the injection well [16,17]; it makes possible to analyze the concentration of mononuclear diatomic molecules (H$_2$, O$_2$, N$_2$) as well as polar molecules (CO$_2$ and CH$_4$) in the water; since the detection of a monoatomic gas such as He is not possible with this type of sensor, only the indirect effect on the concentration of other dissolved gases can be detected;
- A device for water pumping from the aquifer and degassing by mechanical agitation; it is combined with an ALCATEL ASM 122D transportable mass spectrometer to measure the helium concentration in the degassed gas mixture.

## 3. Results

### 3.1. Tracers

The tracers were analyzed using spectrofluorimetry in the CETRAHE lab at the University of Orléans (Table 5). The assay of each of these tracers was performed using a calibration curve established with the same tracer used for the test. It should be noted that the spectral analysis technique, via the characteristic excitation and emission spectra, makes it possible to confirm the presence of these fluorescent tracers even when the concentration is low and close to the detection limit, to avoid any confusion with the natural fluorescence of water. The maximum net concentrations obtained at each piezometer are summarised in Table 6.

**Table 5.** Analytical methods and detection thresholds for the analyzed tracers.

| Tracer | Uranine | Sulforhodamine B | Amino G Acid | Li | Br |
|---|---|---|---|---|---|
| Method | ICP-MS | ICP-MS | ICP-MS | IC | IC |
| DL ($\mu$g·L$^{-1}$) | 0.001 | 0.05 | 0.05 | 1 | 1 |

Legend: ICP-MS = Inductively Coupled Plasma-Mass Spectrometry; IC = Ionic chromatography; DL = Detection Limit).

**Table 6.** Maximum net tracer concentration per piezometer (in $\mu$g·L$^{-1}$).

| Piezometer | Distance * (m) | Uranine (DL = 0.001) | Sulforhodamine (DL = 0.050) | AGA (DL = 0.100) | Lithium (DL = 0.5) | Bromide (DL = 0.5) |
|---|---|---|---|---|---|---|
| PZ1 | −20 | 0.0 | 0.0 | 0.0 | 2.0 * | 2.8 * |
| PZ2 | 0 | 10.2 | 4.1 | 23.9 | 22.0 | 3.1 |
| PZ2BIS | +5 | 19.8 | 21.1 | 30.0 | 148.1 | 45.7 |
| PZ3 | +10 | 1.2 | 0.3 | 0.0 | 3.4 | 5.0 |
| PZ4 | +20 | 0.5 | 0.1 | 0.0 | 2.7 | 1.1 * |
| PZ5 | +30 | 0.3 | 0.0 | 0.0 | 2.0 * | 0.5 * |
| PZ6 | +60 | 0.0 | 0.0 | 0.0 | 1.8 * | 0.9 * |

Legend: Distance = Distance from injection well in the downstream direction (positive values) and upstream direction (negative value), DL = Detection Limit, AGA = Amino G Acid, * background noise.

It thus appears that the PZ1 piezometer (upstream of the injection well) was not reached by the tracer plume and that the PZ2BIS piezometer (5 m downstream) is the only piezometer where the presence of all the tracers was proven. Starting from PZ3, located 10 m downstream of the injection well, some tracers such as the Amino G Acid and the bromide were not detected. Starting from PZ5, located 30 m downstream of the injection well, sulforhodamine B was also no longer detected. At PZ6, the piezometer that is most removed (60 m downstream of the injection well), no tracer was detected in significant concentrations by the end of the monitoring period. Uranine and lithium are the only tracers that were detected in all the piezometers located downstream of the injection point, except at PZ6. These are thus the best-suited tracers for this hydrogeological context, the first one because it is easily detectable in situ (using a field fluorimeter) including at low concentrations (0.1 $\mu$g·L$^{-1}$) and the second one because it proved to be more conservative.

For uranine and lithium, the results were also interpreted using TRAC software [18] considering Fried's analytical solution [19] for the brief injection of a mass of tracer into an infinite volume in flow (Equation (1)):

$$C_{(x,y,t)} = \frac{m}{4\,b\,\pi\,\omega\,t\,\sqrt{D_L\,D_T}} \cdot \exp\left[ -\frac{(x - u\,t)^2}{4\,D_L\,t} - \frac{y^2}{4\,D_T\,t} \right] \tag{1}$$

where $C_{(x,y,t)}$ is the concentration of tracer (kg·m$^{-3}$) at the point with coordinates $(x, y)$ (m) and at time $t$ (s), $m$ is the mass of injected tracer (kg), $b$ the thickness of the aquifer (m), $\omega$ the cinematic porosity (−), $D_L$ the longitudinal dispersion (m$^2$·s$^{-1}$), $D_T$ the transversal dispersion (m$^2$·s$^{-1}$), and $u$ the real flow speed (m·s$^{-1}$). In the context of this test, the fixed parameters are $m = 10^{-3}$ kg, $b = 14$ m and $x$ which corresponds to the distance of each piezometer from the injection well. Note that the y coordinate has been left free, which makes it possible to check whether the piezometers are properly aligned in the main flow axis of the aquifer with respect to the injection well.

At PZ2BIS, 5 m downstream of the injection well, the concentration peak was achieved on the day of injection itself at 11:44 for lithium and at 15:28 for uranine, that is, 1.15 and 4.83 h respectively after the start of injection (Figure 10). Despite this difference in transit time, the hydrodynamic parameters used for the calibration of the breakthrough curves are the same for the two tracers: only the retardation factor varies, being fixed at 1.0 for lithium and 1.6 for uranine. The cinematic porosity is thus equal to $1.40 \times 10^{-2}$ and the permeability $1.10 \times 10^{-3}$ m·s$^{-1}$, values in accordance with those previously obtained in test pumping.

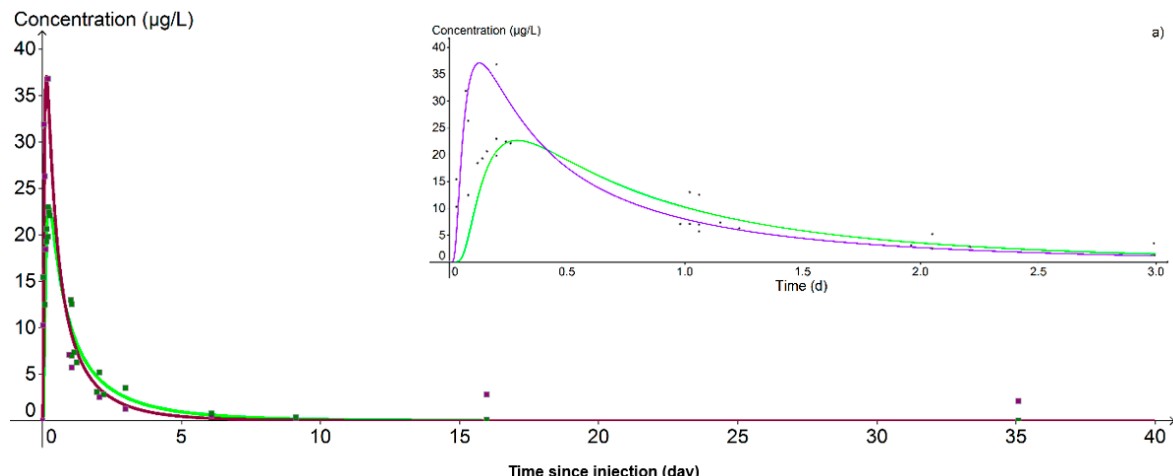

**Figure 10.** Evolution of uranine (green curve) and lithium (purple curve) concentrations at the nearest downstream piezometer (PZ2BIS). (**a**) Detail of the 3 first days. The peak at 148.1 $\mu g \cdot L^{-1}$ obtained with the lithium is not shown because it could not be simulated.

From the PZ3 piezometer, 10 m downstream of the injection well, it is no longer possible to fit the data on a single curve because the recovery is bimodal. The first peak reflects a rapid arrival of the tracer by a preferential path (fissured zone?) while the second peak corresponds to a slower propagation within the matrix aquifer. In Figure 11, two distinct fits were therefore applied to each first peak (dashed curves) and second peak (solid curves). Table 7 shows that the porosity obtained is fairly uniform around the mean value of $5.97 \times 10^{-2}$ $m \cdot s^{-1}$ regardless of the piezometer or tracer studied, but that the permeability varies more strongly around the mean value of $4.11 \times 10^{-3}$ $m \cdot s^{-1}$ depending on the adjustment made. These values are also significantly higher than those obtained at PZ2BIS, which is interpreted as resulting from an environment with multiple porosity, of both matrix and fissure type, once a larger aquifer volume is involved. As before, we observe a faster propagation of the plume at PZ4 (3.9 $m \cdot d^{-1}$) than at PZ3 (1.6 $m \cdot d^{-1}$): however, these speeds are 2 to 3 times lower than during the tracing test carried out in 2012 (10 $m \cdot d^{-1}$ and 3 $m \cdot d^{-1}$, respectively), which seems to be due to a low groundwater table which started exceptionally early this year. It should also be noted that this speed artificially reached 104 $m \cdot d^{-1}$ during the injection, at the PZ2BIS which is a piezometer directly influenced by the injection conditions.

**Table 7.** Average hydrodynamic characteristics resulting from the calibration of the recovery curves.

| Peak | First Peak (With PZ2BIS) | | Second Peak (Without PZ2BIS) | | Average |
|---|---|---|---|---|---|
| **Tracer** | **Lithium** | **Uranine** | **Lithium** | **Uranine** | |
| Porosity (−) | $4.85 \times 10^{-2}$ | $6.35 \times 10^{-2}$ | $5.67 \times 10^{-2}$ | $7.00 \times 10^{-2}$ | $5.97 \times 10^{-2}$ |
| Permeability ($m \cdot s^{-1}$) | $4.47 \times 10^{-3}$ | $1.11 \times 10^{-2}$ | $3.03 \times 10^{-4}$ | $5.86 \times 10^{-3}$ | $4.11 \times 10^{-3}$ |

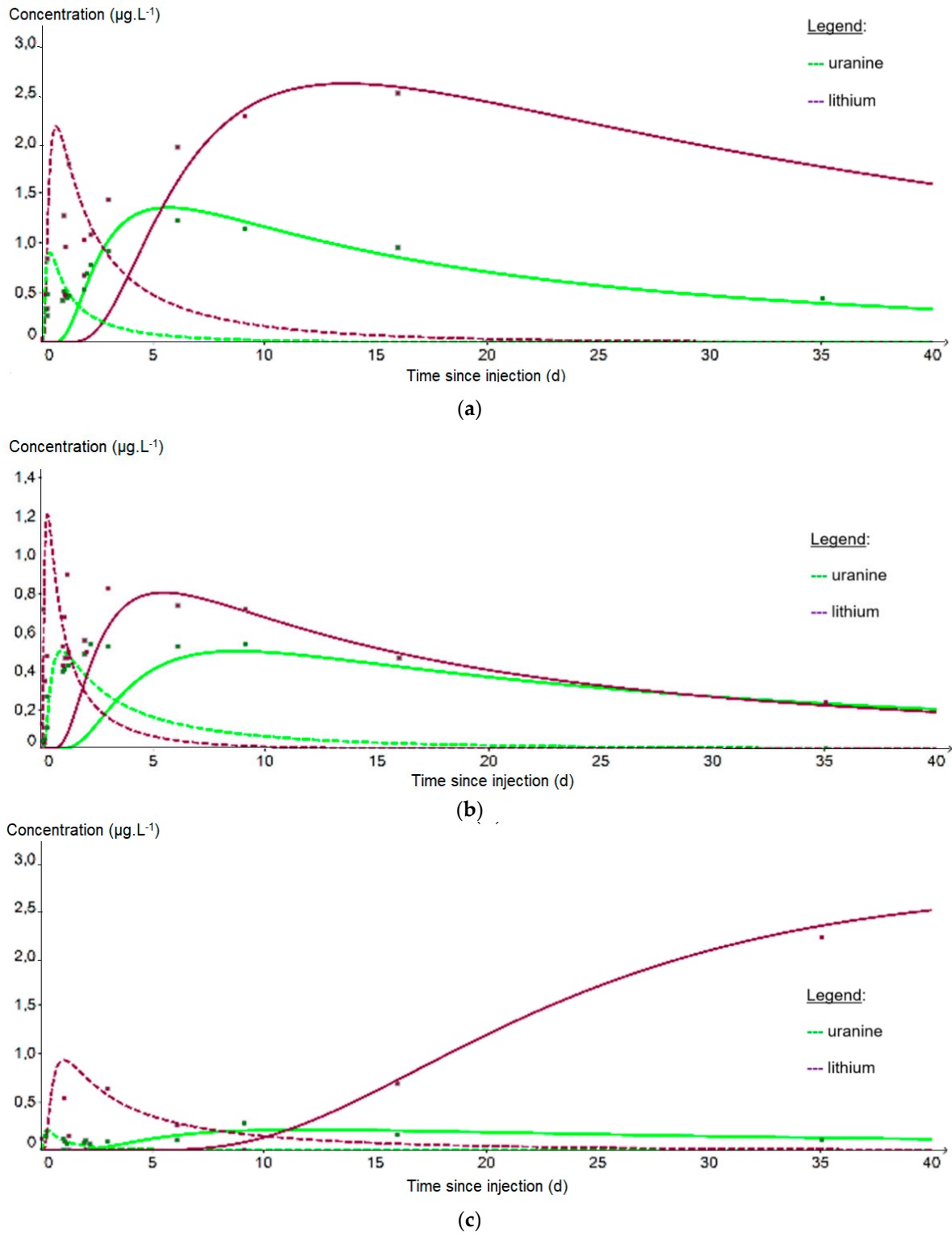

**Figure 11.** Evolution of the tracer concentrations at the downstream piezometers: (**a**) PZ3; (**b**) PZ4; (**c**) PZ5. The curves correspond to the fit of the first peak for the dashed curve and of the second peak for the solid curve; the purple curve represents lithium and the green curve uranine.

*3.2. Dissolved Helium Concentration*

The dissolved helium was extracted from the water sampling vessels by partial degassing by mechanical agitation, after which the extracted gaseous mixture was directly analyzed on site using

an ALCATEL ASM 122D mass spectrometer. The results obtained during baseline measurements at the two reference piezometers, located upstream (PZ1) and far downstream (PZ6), indicate that the groundwater does not contain a significant amount of helium. The measured values of dissolved gas are less than the equilibrium concentration with the surface-atmosphere (helium content about 5 ppm).

During the experiment, the arrival of the helium plume at the PZ2BIS piezometer, 5 m downstream of the injection well, occurred very quickly after injection (Figure 12): the maximum concentration of 1.47 mg·L$^{-1}$ was measured 30 min after injection. After this, the helium concentration in water decreases. It has almost returned to its initial state at this piezometer 40 days after injection. In PZ3 and PZ4 piezometers, located respectively 10 m and 20 m downstream, the dissolved helium concentrations were significantly lower as at PZ2BIS. The arrival of the helium was detected 3 h after the injection at PZ3 piezometer; and a little more than 5 h after at PZ4. At these two piezometers, the maximum helium concentrations were about 3 and 8 µg·L$^{-1}$, respectively, and were recorded 9 days after injection. No significant trace of dissolved helium was measured at the other piezometers located further downstream (PZ5 at 30 m and PZ6 at 60 m).

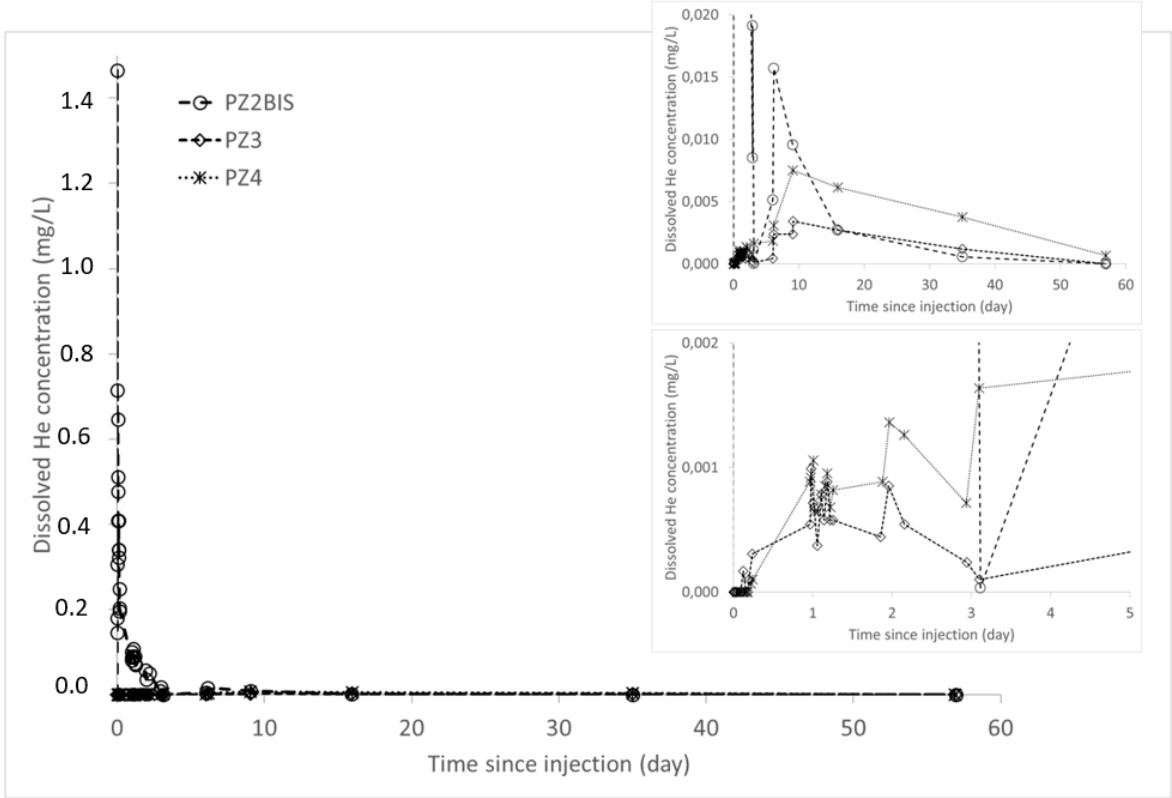

**Figure 12.** Dissolved helium concentrations in the samples taken at the downstream piezometers PZ2BIS, PZ3, and PZ4.

## 4. Discussion

Following this test, we can see that the experimental protocol for saturating water with gas and then injecting it into the groundwater is operational, as is the way of monitoring the saturated zone. However, the results obtained lead us to propose a certain number of improvements for the hydrogen injection experiment.

Concerning the gas saturation of the water in the 5 m$^3$ tank, it will not be possible—for safety reasons—to allow hydrogen to bubble all night to obtain maximal saturation at the time of injection, as was done with the helium. Hydrogen is an easily flammable gas requiring the establishment of an ATEX zone and suitable control measures. These measures are difficult to ensure overnight. As a result, the hydrogen bubbling will have to be interrupted in the evening to resume the next morning. This can

lead to a delay of several hours in reaching an optimum level of hydrogen saturation in the water in the tank and, consequently, the same delay in all the subsequent operations (injection, monitoring measurements, etc.). To increase saturation kinetics, the number of gas outlets at the bottom of the $5 m^3$ tank will be doubled, from 200 to 400.

The first $1 m^3$ tank will hold the fluorescent and ionic tracers that have been shown to provide the best performances: uranine and lithium. Considering the weakness of the signal obtained during this test, owing to a strong dilution of the tracers in the groundwater, they will be used at a concentration higher than an order of magnitude: $10 g \cdot L^{-1}$ instead of $1 g \cdot L^{-1}$. In addition, the water in this tank will also be saturated with helium, to be used as an inert tracer gas to compare its behavior with that of hydrogen, a potentially reactive gas in this aquifer context.

The second $5 m^3$ tank will be saturated with hydrogen by means of bubbling in a gaseous state during the first day of preparing the materials, as well as during the following morning. The injection of the hydrogen-saturated water will therefore take place at the start of the afternoon. In the test conducted with helium, the two tanks were drained successively, but owing to their respective geometries, the injection rate of the second tank was found to be significantly higher ($3.2 m^3 \cdot h^{-1}$) than that of the first ($1.7 m^3 \cdot h^{-1}$). Following this, the two plumes probably coalesced, which hampered the interpretation of the tracer breakthrough curves, and probably diluted the plume of dissolved gas. To avoid this, we will apply a latency time of $\frac{1}{2}$ h between the two injections, and the emptying rate of the second tank containing dissolved hydrogen will be retained at less than or equal to that of the first tank containing the tracers.

The PZ2BIS piezometer placed directly downstream of the point of injection provided the best recovery curves and will thus be considered to be the principal monitoring piezometer. As such, given the speed of the response obtained (1.15 h), it must be equipped with a specific monitoring device to provide continuous data acquisition: dissolved hydrogen measurement probe, physicochemical measurement probe, and borehole fluorimeter. This equipment must be available in duplicate to be able to monitor the other piezometers manually. As soon as the tracer signal has disappeared from the PZ2BIS, the continuously recording borehole fluorimeter will be moved to the piezometers located further downstream (PZ3, PZ4, and PZ5).

The piezometers must not all be sampled at the same frequency, but at specific time intervals in function to their distance from the injection well, and the current hydrogeological conditions, by taking particular account of the propagation speed of the fluorescent tracer. During the current test, the duration of monitoring (40 days) did not permit a satisfactory sampling of the most distant piezometers, namely PZ5 (30 m downstream) and PZ6 (60 m downstream). This duration will therefore be significantly increased, but at a rate of only one sample per week from the 5th week of monitoring: the total duration of the monitoring may vary from 60 to 80 days depending on the hydrogeological conditions at the time (high or low water). It is, however, suggested that a period of high water is favored to reduce the monitoring time. In all cases, only the PZ2TER will operate continuously for the measurement of dissolved gases using Raman and IR spectrometers ($O_2$, $N_2$, $H_2$, $CO_2$, and $CH_4$).

Finally, the piezometry of the aquifer will be measured twice a day at all piezometers during the week of injection, and then once a day thereafter, to detect any variation in the speed or flowing direction of the aquifer. An automatic water depth measurement probe will also be placed at the bottom of PZ2 to measure the amplitude of the piezometric dome induced by the injection.

## 5. Conclusions

A test of the combined injection of tracers (organic and ionic) and helium-saturated water was done in April 2019 to assess and optimize the concept of injecting water saturated with hydrogen, planned for later, and monitoring its physicochemical properties.

The test has confirmed the technical feasibility, under field conditions, of saturating a significant quantity of water with a low-solubility gas and injecting it in a controlled manner into a shallow aquifer.

It was possible to properly monitor the propagation of the dissolved gas plume in the aquifer with the means of analysis used. Helium could be detected up to 20 m downstream of the injection well by mass spectrometer analysis of the gas mixture obtained through partial degassing of water samples by mechanical agitation.

Among the five tracers used, uranine and lithium were shown to be the most effective. The first is a colored fluorescent organic tracer, easily and continuously detectable in situ but affected in this specific hydrogeological context (fine matrix and fissure porosity) by a certain retardation factor with respect to the propagation of the water. The second is a colorless ionic tracer, not affected by such a retardation factor, but not detectable in situ. To obtain a cleaner signal, the tracer mass used will be ten times higher when injecting the water saturated with hydrogen.

The temporal modeling of the post-injection evolution of these two tracers reveals, as a function of the distance from the injection well, two distinct hydrodynamic regimes linked to the existence of a multiple porosity, of both matrix and fissure type. These elements will be essential in understanding the transport of the hydrogen plume during the next injection simulation.

The preparation and the conditions of injection of the tracer tank and hydrogen-saturated water tank have been modified to take the results obtained into account: doubling of the number of bubbling outlets in the 5 m$^3$ tank bubbling device, establishment of a latency period between the two injections, reduction of the flow rate of the second tank.

Finally, the protocol of monitoring has also been modified: the establishment of specific monitoring of the PZ2BIS piezometer with continuous in situ recordings of a maximum of data, adaptation of the sampling schedule to the specificity of each piezometer and increase in the overall monitoring time.

Thus, the adoption of all of these improvements will permit proper execution of the main experiment of injecting hydrogen-saturated water and carrying out the associated monitoring, which will also be preferentially done during periods of high water.

**Author Contributions:** Conceptualization, methodology, and validation, P.G., E.L., S.L., and Z.P.; analysis, S.L., P.G., Z.P., E.L., P.d.D., and N.J.; writing—original draft preparation, S.L., P.G., and Z.P.; writing—review and editing, S.L., P.G., Z.P., E.L., P.d.D., and N.J.; supervision, P.G. All authors have read and agreed to the published version of the manuscript.

**Funding:** This research was funded by the French Scientific Interest Group GEODENERGIES in the framework of the ROSTOCK-H project (Risks and Opportunities of the Geological Storage of Hydrogen in Salt Caverns in France and Europe).

**Conflicts of Interest:** The authors declare no conflict of interest. The funders had no role in the design of the study; in the collection, analyses, or interpretation of data; in the writing of the manuscript, or in the decision to publish the results.

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
