# Peer review of "Monitoring Scheme for the Detection of Hydrogen Leakage from a Deep Underground Storage. Part 1: On-Site Validation of an Experimental Protocol via the Combined Injection of Helium and Tracers into an Aquifer"

_applsci, doi:10.3390/app10176058_

Round 1

Reviewer 1 Report

The paper presented by Lafortune et al. investigated on the underground storage hydrogen leakage using an on site experimental protocol. With a combined injection of helium and tracers (organic and ionic) into the aquifer the Authors was able to simulate the behaviour of the possible hydrogen leakage plume. During the experiments the Authors reconstruct the propagation of the dissolved gas plume into the aquifer choosing the most suitable tracer: uranine and lithium that best fit a suitable protocol for future helium injections.

The paper is well organized and the English is fluent and smoothly.

References must be updated.

Several times french language substitute the english (refer to attached file "line by line comments")

The figures and graphs require a substantial re-progettation. several times are not clearly legible and the graphs are not self-explanatory.

To best investigate the evolution of a plume with a multi-temporal approach it is fundamental to evaluate shape and volume of the plume during time. For this reasons geostatistics represent, without any doubt, the best way to simulate the diffusion and propagation of a chemicals drived by groundwater dynamics. Authors are kindly invited to simulate plume behaviours using geostatistical maps (i.e. kriging contour maps). A new paragraph must be added to the text with methodology and results of the geostatistical simulations. To facilitate this modifications Authors could refer to the following papers:

1) Rossi et al. 2019 - Groundwater Autochthonous Microbial Communities as Tracers of Anthropogenic Pressure Impacts: Example from a Municipal Waste Treatment Plant (Latium, Italy). Water 2019, 11, 1933; doi:10.3390/w11091933

2) Frollini et al. 2019 - A proposal for groundwater sampling guidelines: application to a case study in southern Latium. Rend. Online Soc. Geol. It., Vol. 47 (2019), pp. 46-51. (https://doi.org/10.3301/ROL.2019.09)

Overall, this work represent a good contribution and deserves to be published after major revisions.

LINE by LINE comments:

LINE 76 - HYDROGEOLOGICAL TRACERS. Why the Authors are using this term ? simply TRACERS is enough. Please remove the term hydrogeological from the whole manuscript when linked to TRACERS.

LINE 81 - "before reaching the surface." References are needed.

LINE 90 - "path of migration to the surface". References are needed.

LINE 104 - "The literature shows that". References are needed.

LINE 112 -  site map is required showing the location of the site and its coordinates. (a geological map it is also important)

LINE 115 - Wrong orientation!. Drainage direction must be expressed using opposite coordinates such as NW-SE, E-W, WSW-ENE etc. where the second term express the verse. For example groundwater drainage from East to West is expressed as E-W.

FIGURE 1 - A detailed geological cross section it is required, with symbols and bedding of the strata. Fig. A must be removed while reference distance line must be added in the lower edge of the cross section.

FIGURE 2 - Figure 2 it is unusefull to the paper and must be substituted with geostatistical contour maps to better understand the concentration and distribution of the elements measured.

FIGURE 3 - same comments as for Figure 2

LINE 172 - "TABLEAU 4" change it in english. It is very frequent in the text to find different language terms. Please pay attention and substitute the whole wrong language terms.

LINE 181 - same as LINE 172.

FIGURE 4 - Be sure to use only one code: 1/2 or a/b. Figure 4(2/b) is too dark and should be lightened to improve it.

FIGURE 5 - same as for Figure 4.

LINE 277 - again "Tableau 5" refer to comment LINE 172.

FIGURE 6 - Graph must me modified to better increase the legibility. in particular horizontal range must be changed.

PLEASE REFER TO THE ATTACHED FILE (line-by-line.pdf) FOR FURTHER COMMENTS

Reviewer 2 Report

The paper present interesting experiments about an helium injection in an aquifer in order to prepare hydrogen injection.

This experiment is worth to be published however it could be also meaningfull to link this experiment to the ongoing researches about H2 in subsurface : native H2 as H2 produced by in situ combustion. There is no reference to the fact that H2 exists in subsurface in this paper, a lot of developments have been done to detect and monitore native H2 for 10 years and its look strange to don't take advantage of these results.

Why for instance the authors are choosing to monitore the disolved H2 and not the H2 in the soil ? (see for instance Larin et al., 2015 and Prinzhofer et al. 2019 for H2 soil concentration measurements). It has to be discussed.

Bacterial activity has a huge influence of the H2 consumption in the soil (see myagkiy et al., 2019) even in couple of hours bacteries may consume a lot of H2. In underground storage the bacterial activity may change the gas chemical signature (see Ranchou et al., 2020), it is not the case for Helium so the similarities between the two gases have some limitations that have to be discussed in the paper. Obviously the H2 is also chemically reactive an can be consume or absord in the clay rich rock (Truche et al. 2018).

The discussion about the requested time for the bubbling of H2 may be for instance reduced taking advantage of the fact that the H2 solubility is strongly increasing with the pressure (see Lopez et al., 2019).

So globally I suggest to do the link between this experiment and the other researchs around H2 in subsurface. It will be much more interesting for the reader and may be the authors will propose another protocol based on the comparison. 

Two last points

1-concerning the abstract: it looks as an introduction, there is no result. It has to be improved

2- concerning the P2G2P, a lot of analysts pointed out the fact that it don't have any economic sens, up to now, to do it since the efficiency of electrolysers and fuel cell are low. So since it is not the core of the paper it is better to stay neutral about this topic. We may need H2 long term storage even without doing P2G2P.

Round 2

Reviewer 2 Report

much better